# The effectiveness of early start of Grade III response to dengue in Guangzhou, China: A population-based interrupted time-series study

Li Li[1☯], Wen-Hui Liu[1,2☯], Zhou-Bin Zhang[2☯], Yuan Liu[2], Xiao-Guang Chen[3], Lei Luo[2]*, Chun-Quan Ou[1]*

**1** State Key Laboratory of Organ Failure Research, Department of Biostatistics, Guangdong Provincial Key Laboratory of Tropical Disease Research, School of Public Health, Southern Medical University, Guangzhou, China, **2** Guangzhou Center for Disease Control and Prevention, Guangzhou, Guangdong, China, **3** Department of Pathogen Biology, Key Laboratory of Prevention and Control for Emerging Infectious Diseases of Guangdong Higher Institutes, Guangdong Provincial Key Laboratory of Tropical Disease Research, School of Public Health, Southern Medical University, Guangzhou, China

☯ These authors contributed equally to this work.
* llyeyq@163.com (LL); ouchunquan@hotmail.com (CQO)

**Data Availability Statement:** The data that support the findings of this study are available from Guangzhou Center for Disease Control and

## Abstract

In 2019, dengue incidences increased dramatically in many countries. However, the prospective growth in dengue incidence did not occur in Guangzhou, China. We examined the effectiveness of early start of Grade III response to dengue in Guangzhou. We extracted the data on daily number of dengue cases during 2017–2019 in Guangzhou and weekly data for Foshan and Zhongshan from the China National Notifiable Disease Reporting System, while the data on weekly number of positive ovitraps for adult and larval *Aedes albopictus* were obtained from Guangzhou Center for Disease Control and Prevention. We estimated the number of dengue cases prevented by bringing forward the starting time of Grade III response from September in 2017–2018 to August in 2019 in Guangzhou using a quasi-Poisson regression model and applied the Baron and Kenny's approach to explore whether mosquito vector density was a mediator of the protective benefit. In Guangzhou, early start of Grade III response was associated with a decline in dengue incidence (relative risk [*RR*]: 0.54, 95% confidence interval [CI]: 0.43–0.70), with 987 (95% CI: 521–1,593) cases averted in 2019. The rate of positive ovitraps also significantly declined (*RR*: 0.64, 95% CI: 0.53–0.77). Moreover, both mosquito vector density and early start of Grade III response was significantly associated with dengue incidence after adjustment for each other. By comparing with the incidence in Foshan and Zhongshan where the Grade III response has not been taken, benefits from the response starting in August were confirmed but not if starting from September. Early start of Grade III response has effectively mitigated the dengue burden in Guangzhou, China, which might be partially through reducing the mosquito vector density. Our findings have important public health implications for development and implementation of dengue control interventions for Guangzhou and other locations with dengue epidemics.

Prevention but restrictions apply to the availability of these data, and so the data are not publicly available. Permission can be requested by contacting Guangzhou Center for Disease Control and Prevention.

**Funding:** This work was supported by National Nature Science Foundation of China to CQO [81973140], China Postdoctoral Science Foundation to L LI [2020M672744] and National Institutes of Health, USA to XGC [AI 136850]. The funders had no role in study design, data collection and analysis, decision to publish, or preparation of the manuscript.

**Competing interests:** The authors have declared that no competing interests exist.

## Author summary

There is a lack of data on comparing the observed dengue incidences under the real-world scenarios that interventions commenced at different times. In 2019, WHO scaled up the response to dengue due to the escalation of outbreaks occurring in many countries. In the same year, local government in Guangzhou started the Grade III response to dengue one month ahead in August. It is uncertain the degree to which the early intervention mitigated dengue burden. Our study examined the effectiveness of early start of Grade III response in Guangzhou using a quasi-Poisson regression model by comparing the dengue incidence with early start of Grade III response and that under the counterfactual scenario that the Grade III response began in September as in 2017 and 2018. We estimated that 987 dengue cases were averted due to the early start of Grade III response, which were equivalent to 71.4% of the total number of local dengue cases in 2019. Early start of Grade III response reduced the dengue burden, which might be partially through controlling the mosquito vector density. Dengue intervention strategies applied in Guangzhou could provide experience on how to effectively prevent and control dengue for other locations with dengue epidemics.

## Introduction

Dengue is the most rapidly spreading mosquito-borne disease throughout the world and it is currently endemic in 128 countries [1]. Globally, around half of the population are at risk of dengue virus infection, with approximately 400 million people infected by dengue viruses annually [2]. Climate change, population growth, increasing human mobility, and unplanned urbanization are expected to facilitate the spread of dengue viruses and thereby aggravate the dengue disease burden [3–6]. In 2019, World health Organization (WHO) listed dengue virus as the world's top 10 public health threats [1].

Vaccination is an intervention for the prevention of dengue infection. However, dengue vaccination programs are only recommended for areas with a high dengue burden, considering the vaccine safety issues for seronegative individuals and low vaccine efficacy [7–9]. Even as effective vaccines become available, vector control is crucial in dengue prevention and mitigation [10, 11]. Previous studies have reported the effectiveness of interventions which included vector control measures in locations such as two Chinese cities of Guangzhou [12, 13] and Ningbo [14], Sri Lanka [15], and Singapore [16]. In addition, Liu et al. illustrated that the implementation of early rigorous interventions could reduce dengue epidemic through simulating dengue incidences under different scenarios [17]. However, there is a lack of data on comparing the effectiveness of interventions commenced at different times for the identification of appropriate start time of the intervention.

Dengue disease burden distributes disproportionately globally, with Asia representing 75% of the burden [18]. Guangzhou, located in the subtropical climatic zone (latitude: 23˚07′N; longitude 113˚15′E) and having a permanent population of 14.9 million, is most highly influenced by dengue in China. During 2006–2014, approximately 70% of the dengue cases in China were reported in Guangzhou [19]. During 2017–2018, the peak dengue season was from August to November. Guangzhou Municipal Health and Family Planning Commission (GZ MHFPC) divided the events related to dengue into six grades and formulated a series of responses for events of different grades. Considering the dengue epidemic, funding and manpower required for the initiation, the local government started the Grade III response which

targeted the Grade III events related to dengue in September in 2017 and 2018 to curb the proliferation of dengue transmission in the population. However, the initiation of Grade III response mainly triggered by the event may be not timely for adequately controlling the consequent epidemics.

In 2019, large dengue outbreaks occurred in many locations and especially, places such as Bangladesh, Pakistan, and WHO Region of the Americas witnessed their worst dengue outbreaks in recent years [1, 20–22]. Due to the escalation of outbreaks, WHO scaled up the global response to dengue [1]. In Guangzhou, a higher mosquito vector density in early months was a potential signal of a large dengue epidemic in the year [23]. In response to the high risk of increasing severity of dengue epidemic, the start time of Grade III response was advanced in August. This provided us a precious opportunity to examine the impact of early implementation of control program on the additional reduction in dengue disease burden. The present study aims to 1) gauge the effect of early start of Grade III response on dengue incidence and 2) explore the role of vector control in dengue prevention and mitigation, when the beginning time of Grade III response was advanced one month ahead in 2019 in Guangzhou, China.

## Methods

### Data source

We extracted the data on daily number of dengue cases confirmed by serological and/or virologic methods in Guangzhou during 2017–2019 and weekly data for Foshan and Zhongshan from the China National Notifiable Disease Reporting System. The dengue cases were classified into local cases and imported cases. The mosq-ovitrap is the most widely employed device for the surveillance of *Aedes albopictus* [24, 25] and it is currently used for routine surveillance of container-breeding *Aedes* in 172 streets in all of the 11 districts of Guangzhou. Data on the number of positive ovitraps for adult and larval *Aedes albopictus* and total number of retrieved traps were obtained from Guangzhou Center for Disease Control and Prevention (GZ CDC). Data collection frequency was determined according to dengue epidemic, from once a month between January and March, to once a week between July and November. Mosquito ovitrap index (MOI) was calculated as the number of positive ovitraps for adult and larval *Aedes albopictus* per 100 traps which were retrieved [26]. Daily meteorological data on ambient mean temperature, relative humidity, and rainfall were downloaded from the China Meteorological Data Sharing Service System [27].

### The Grade III response to dengue

GZ MHFPC formulated a series of responses for the events related to dengue, which were divided into six grades (i.e., VI, V, IV, III, II, I) according to the severity of events, with the Grade I events being the most severe ones (S1 Text) [28]. Generally speaking, the responses is comprised mainly of vector surveillance, vector control, raising public awareness of the health risk related to mosquito vectors, public announcement of the outbreak. The most important counterpart of these responses is vector control, including killing adult mosquito vectors, clearing indoor and outdoor mosquito breeding grounds. The core area is defined as the area within a radius of 200 meters from a dengue case, while the warning zone is defined as the area within a radius of 400 meters but outside a radius of 200 meters of a dengue case. Streets are required to kill adult mosquito vectors indoors and outdoors in core areas within three days. Subsequently, the intervention is implemented every three days for three times and then once a week until the end of the epidemic. The intervention is meanwhile conducted in warning zones with different frequency (i.e. once a week if mosquito vector density is reduced to the

safety level; every three days if mosquito vector density still exceeds the safety level. Details on the safety level are provided in S2 Text).

Vector control requires efforts from government, communities and individuals. Interventions formulated for the events of six grades related to dengue are in general consistent, but the degree of government involvement differs. The responses for events of Grade VI-IV rely mainly on the efforts from local health authority with limited resources. However, local government dominates the work of dengue control when Grade III-I responses start. The aims of the Grade III response, which targets the Grade III events related to dengue (S1 Text), are quickly controlling dengue outbreak and preventing rapid spread of dengue in the population. The initiation of Grade III response is influenced by various factors, such as dengue epidemic, funding and manpower required for the initiation, national and global events occurring at the time. After commencing the Grade III response, the intervention for dengue control is intensified, additional works are required to be conducted, more resources are allocated and the cross-departmental collaboration is reinforced, with direct supervision of senior officials.

## Data analysis

Since the data on mosquito vector density were collected at the end of month during January and March, the weekly data were available only for the last weeks of these months, while the values of mosquito vector density were missing in other weeks of these months and the missing values were not imputed. The values of rainfall were highly right skewed (more than half of the values were zeroes) and there were a few outlier values (S1 Fig). The inclusion of rainfall as a continuous variable in the model would impair the inference for the association between rainfall and dengue incidence. Thus, we categorized rainfall into three levels (i.e. 0, 0.05–3.85, >3.85mm; 0.05 and 3.85mm were the minimum and median of the non-zero values of rainfall). We compared the effect of Grade III response commencing in August 2019 on local dengue incidence with the counterfactual scenario that the Grade III response started in September as usual by associating an indicator variable of the Grade III response in 2019 and daily incidence of dengue in Guangzhou using a quasi-Poisson regression model [15]. The model evaluating the effect of early start of Grade III response on dengue incidence (Model 1) was as follows:

$$
\begin{aligned}
\log(E[D_t]) = &\ \alpha_D + \textit{offset}(\log[Pop_t]) + ns(Day_t, df = 4) + \gamma_D Year_t \\
&+ \theta_{Temp} Temp_{t,0-22} + \theta_{Hum} Hum_{t,0-69} + \beta_D Int_{t,10-21}
\end{aligned}
\tag{1}
$$

where $t$ is sequential days from January 1, 2017 to December 31, 2019; $D_t$ and $Pop_t$ are daily number of local dengue cases and population at the time point $t$, respectively. We included the logarithm transformation of population into the model and constrained the corresponding regression coefficient to be one. A natural cubic spline function $ns(.)$ with four degrees of freedom ($df$s) was used for the calendar day in each year to control for the seasonality of dengue incidence. In addition, a linear function of calendar year was also included. Candidate climatic variables included daily mean temperature, relative humidity, and a categorical variable of rainfall. We selected climatic variables and corresponding maximum lags according to the Quasi Akaike Information Criterion (QAIC). In Model 1, $Temp_{t,0-22}$ and $Hum_{t,0-69}$ are metrics constructed by applying the distributed lag non-linear model (DLNM) to temperature and relative humidity, with maximum lags of 22 and 69 days, respectively (S1 Table). We constrained the time lag for the indicator variable of the Grade III response in 2019 (1: between August 9 and December 18, 2019; 0: otherwise) to be at least 10 days, considering the intrinsic incubation period (IIP) and time delay from illness onset to reporting cases, and selected the maximum lag based on QAIC. The cross-basis of an indicator variable of the Grade III response in

2019 and time lag ($Int_{t,10-21}$) was subsequently included in the model (S2 Table). The cross-basis can indicate the additional effect of early start of Grade III response, since the Grade III response started in September in 2017 and 2018, but in August in 2019 (S3 Text).

We hypothesized that the Grade III response starting one month ahead in August could influence local dengue incidence through affecting mosquito vector density. To explore the mediator effect of mosquito vector density, quasi-Poisson regression models were further applied to evaluate the effect of early start of Grade III response on the rate of positive ovitraps for adult and larval *Aedes albopictus* (Model 2) and to associate the indicator variable of the Grade III response in 2019, logarithm transformation of MOI, and weekly number of local dengue cases (Model 3), according to the prerequisites for establishing mediation proposed by Baron and Kenny [29, 30] (S2 Fig). Model 2 and Model 3 were as follows:

$$
\begin{aligned}
\log(E[Pos_{tw}]) = \; & \alpha + offset(\log[Trap_{tw}]) + ns(Week_{tw}, df = 4) \\
& + \gamma Year_{tw} + ns(Temp_{tw}, df = 3) + ns(Hum\_0 - 7_{tw}, df = 3) \\
& + \beta Int_{tw}
\end{aligned}
\tag{2}
$$

$$
\begin{aligned}
\log(E[D_{tw}]) = \; & \alpha_{Dw} + offset(\log[Pop_{tw}]) + ns(Week_{tw}, df = 4) \\
& + \gamma_{Dw} Year_{tw} + ns(Temp\_0 - 3_{tw}, df = 3) \\
& + ns(Hum\_0 - 10_{tw}, df = 3) \\
& + \delta \log(MOI\_1 - 3_{tw}) + \beta_{Dw} Int_{tw-3}
\end{aligned}
\tag{3}
$$

where $tw$ is sequential weeks from 1 to 156; $Pos_{tw}$ and $Trap_{tw}$ are the number of positive ovitraps for adult and larval *Aedes albopictus* and total number of traps at the time point $tw$, respectively; $Week_{tw}$ represents the calendar week in each year; $D_{tw}$ and $Pop_{tw}$ are the number of local dengue cases and population at the time point $tw$, respectively. According to Model 1, moving averages of 0–3 and 0–10 weeks were applied to temperature and relative humidity for Model 3, respectively. The time lags between climatic variables and the positive rate in Model 2 and the lag between the logarithm transformation of MOI and dengue incidence in Model 3 were selected jointly, assuming that climatic factors could influence dengue incidence through mosquito vector density. The current-week temperature and a moving average of relative humidity at a lag of 0–7 weeks were included in Model 2, meanwhile the logarithm transformation of MOI at a lag of 1–3 weeks were included in Model 3. In addition, to describe the effects of early start of Grade III response on mosquito vector density and weekly dengue incidence, the indicator variable of the Grade III response in 2019 of current week and at a lag of three weeks were included in Model 2 and Model 3, respectively according to QAIC (S4 Text and S3 Fig).

Subsequently, we used Foshan and Zhongshan, where Grade III response did not initiate during 2017–2019, as negative controls to examine the effects of Grade III response respectively starting in August and September on dengue incidence. The model (Model 4) was as follow:

$$
\begin{aligned}
\log(E[D_{city,tw}]) = \; & \alpha_{Dw\_m} + offset(\log[Pop_{city,tw}]) + ns(Week_{tw}, df = 4) \\
& + \gamma_{Dw\_m} Year_{tw} + ns(Temp\_0 - 3_{city,tw}, df = 3) \\
& + ns(Hum\_0 - 10_{city,tw}, df = 3) \\
& + \beta_{Dw\_m} Int\_m_{city,tw-3} + \beta_{GDP} GDP_{city} \\
& + \beta_{Green} Green_{city} + \alpha_{city}
\end{aligned}
\tag{4}
$$

where $Int\_m_{city,tw-3}$ is a categorical variable indicating the Grade III response starting in August (August 9–December 18, 2019) and starting in September (September 13–December 19, 2017 and September 14–December 17, 2018) in Guangzhou and no Grade III response in Zhongshan and Foshan during the whole study period; $GDP_{city}$ and $Green_{city}$ represent gross domestic product and areas of green spaces per capita (in parks); $\alpha_{city}$ is a random-effect intercept for three cities.

The reduction in daily number of dengue cases due to early start of Grade III response was estimated by subtracting the predicted number of dengue cases given the Grade III response commenced in August 2019 from that under the counterfactual scenario that the Grade III response started in September 2019. Subsequently, the total number of dengue cases averted was estimated by summing up the daily number of avoided cases during the rest period of 2019 after the Grade III response. To investigate the appropriateness of the models used in the main analysis, histograms of residuals, scatter plots of residuals versus predicted values of response variables, and plots of partial autocorrelation function (PACF) of residuals of the fitted models are presented.

We conducted sensitivity analysis to assess the cumulative relative risk (*RR*) of dengue over 0–21 days associated with early start of Grade III response, checking whether the effect of early start of Grade III response began within 10 days; to check the robustness of the effect estimates of early start of Grade III response and MOI, by changing the parameters and adjusting for the number of imported cases in the models (S5 Text). We conducted all analyses using R software version 3.6.2 (R Foundation for Statistical Computing).

## Results

During 2017–2019, there were 3,871 dengue cases reported in Guangzhou, among which 3,454 (89.2%) were local cases (2017: 873; 2018: 1,199; 2019: 1,382), while 417 (10.8%) were imported cases. The first local case in each year under study was consistently identified in June, after which the number of dengue cases increased until October and then declined (Fig 1 and S4 Fig). MOI, temperature, relative humidity, and rainfall during the months between January and June were on average higher in 2019 than in 2017 and 2018 (S4 Fig).

It was observed that dengue incidence in general increased with temperature and relative humidity, although the associations were not linear (S5 Fig). We estimated that the *RR* of dengue associated with early start of Grade III response in 2019 was 0.54 (95% confidence interval [CI]: 0.43–0.70) over 10–21 days (Fig 2). Accordingly, bringing forward the start time of Grade III response in 2019 prevented 987 (95% CI: 521–1,593) dengue cases (Fig 3). Early start of Grade III response was also associated with a reduction in the positive rate of ovitraps, with a *RR* of 0.64 (95% CI: 0.53–0.77). Mosquito vector density might be a mediator between early start of Grade III response and dengue incidence, given (1) the statistically significant effects of early start of Grade III response on dengue incidence and the positive rate of ovitraps, indicated in Model 1 and Model 2, respectively; and (2) the association between the logarithm transformation of MOI and dengue incidence after controlling for the indicator variable of the Grade III response revealed in Model 3 (*RR*: 1.36, 95% CI: 1.01–1.84) (Table 1). The effect of early start of Grade III response on dengue incidence reduced slightly but remained statistically significant after controlling for mosquito vector density (*RR*: 0.68, 95% CI: 0.52–0.90). The findings implied that early start of Grade III response may reduce mosquito vector density which might indirectly curb the transmission of dengue viruses. In addition, early start of Grade III response could reduce the number of dengue cases in other ways. Results of Model 4 indicated that the Grade III responses starting in August were associated with a decline in dengue incidence, with *RR* of 0.59 (95% CI: 0.45–0.78) but the benefit was not statistically

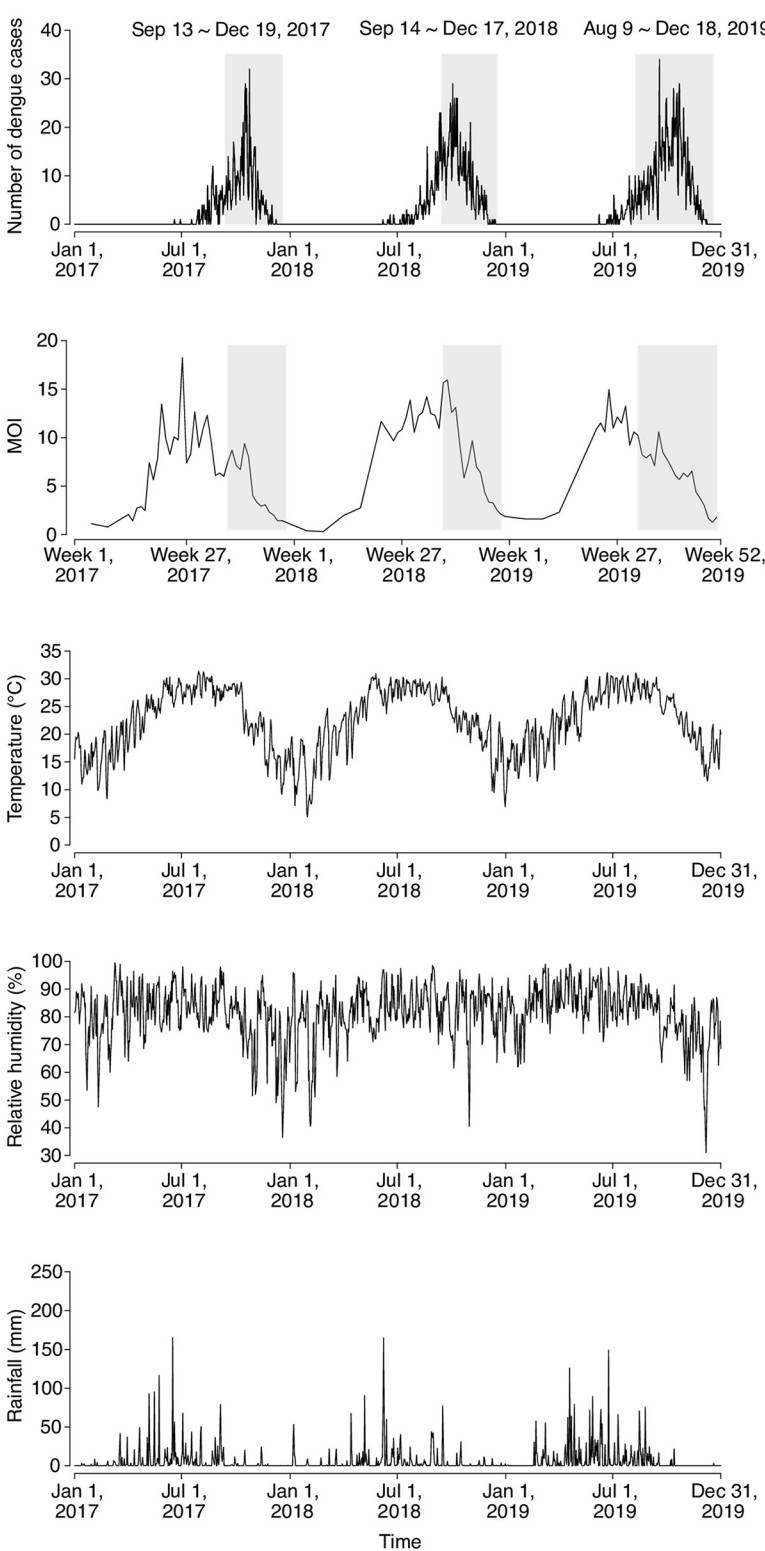

**Fig 1. Time-series of daily number of reported dengue cases, weekly mosquito ovitrap index (MOI), daily temperature, relative humidity, and rainfall in Guangzhou, 2017–2019.** Grey regions indicate the time periods of Grade III response.

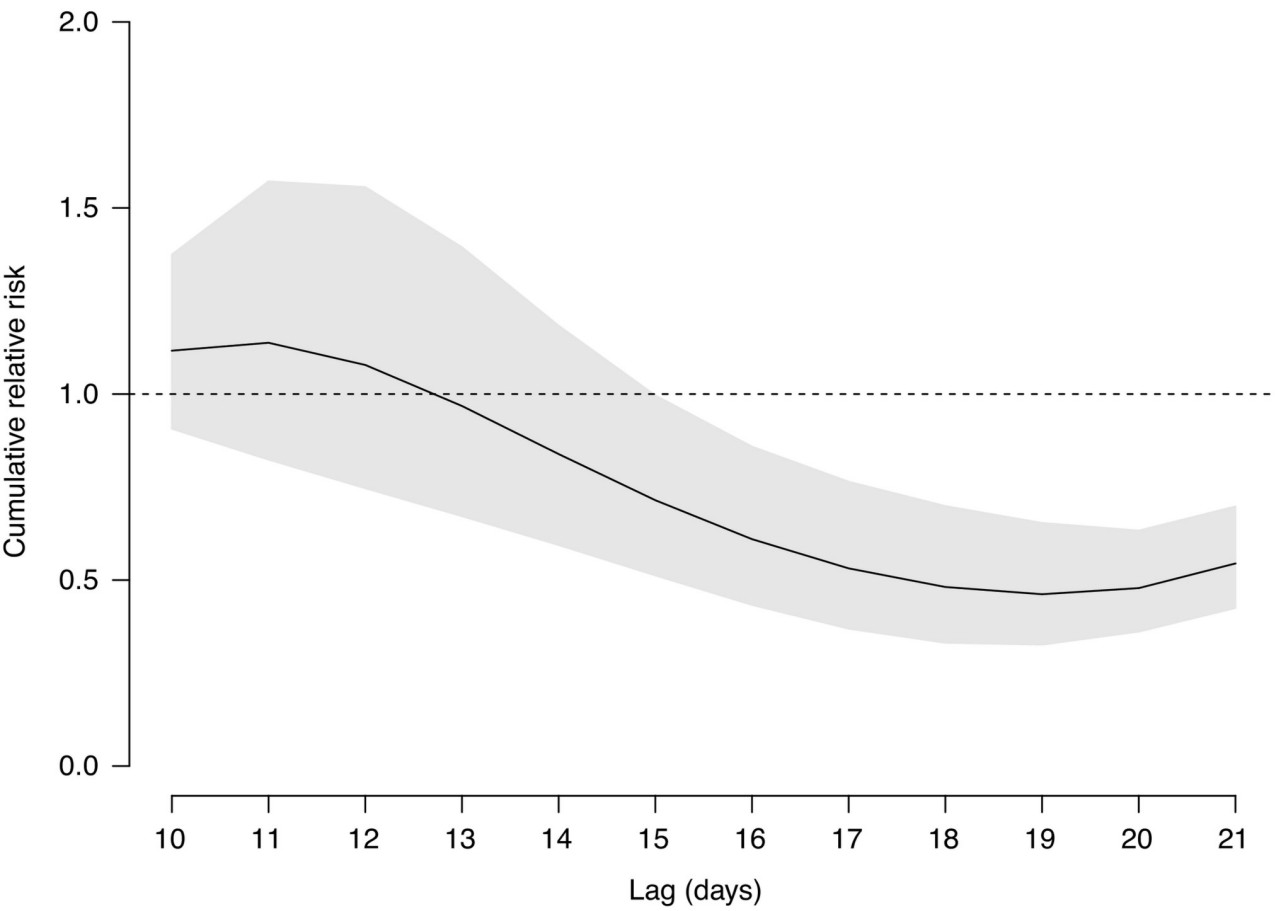

**Fig 2. Cumulative relative risk of dengue due to bringing forward the starting time of Grade III response from September to August in 2019 in Guangzhou over 10–21 days.** The grey region indicates the corresponding 95% confidence interval.

significant if the response started in September (*RR*: 0.91, 95% CI: 0.74–1.12) when comparing with Foshan and Zhongshan where Grade III response was not taken during the whole study period.

Results of model diagnostics suggested that residuals of models were nearly normally distributed and that there was no significant autocorrelation in residuals (S6 Fig). Sensitivity analysis indicated that the results were robust to the specification of parameters and whether the number of imported cases was adjusted or not (S5 Text, S7 Fig, S3, S4 and S5 Tables).

## Discussion

In this study, we assessed the effectiveness of early start of Grade III response in 2019 on dengue control in Guangzhou. In the early months of 2019 between January and June, temperature, relative humidity, MOI, and the number of reported dengue cases in Guangzhou were higher than that in 2017 and 2018 (S4 Fig), signaling that the dengue outbreak was likely to be more severe in 2019 than that in previous two years given the positive associations between these factors and dengue incidence reported in previous studies [23, 31]. However, the prospective increase in dengue incidence did not occur in Guangzhou, which could be attributed to bringing forward the start time of Grade III response from September to August. We estimated that early start of Grade III response in 2019 prevented 987 dengue cases in Guangzhou,

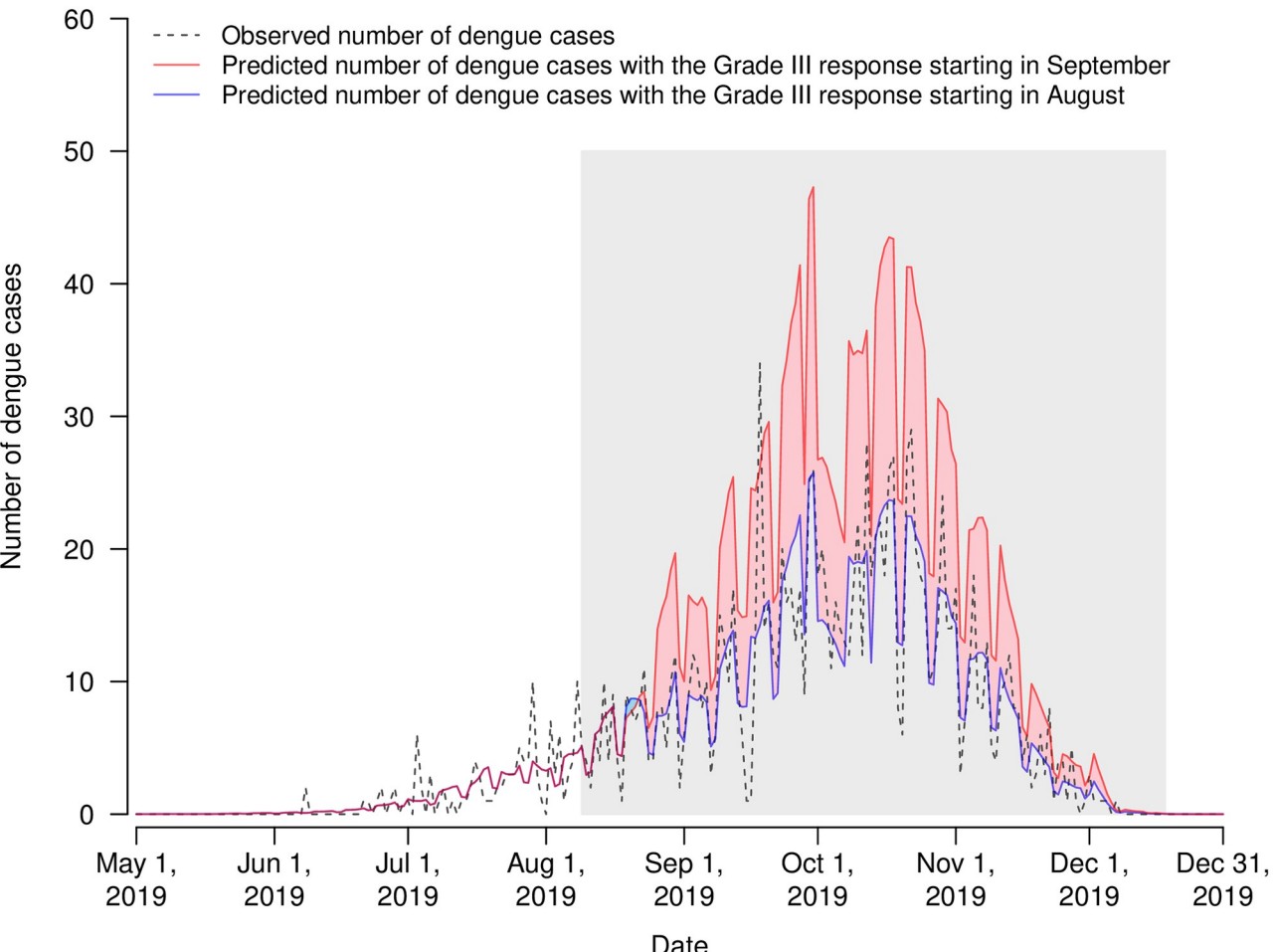

**Fig 3. Observed and predicted numbers of dengue cases with the Grade III response starting in September and August 2019 in Guangzhou.** The grey region indicates the time period of Grade III response in 2019. Blue and red regions represent the subtraction of the predicted number of dengue cases given the Grade III response commenced in August 2019 from the predicted number of dengue cases under the counterfactual scenario that the Grade III response started in September 2019.

which were equivalent to 71.4% of the total number of reported local dengue cases in the year. It was estimated that approximately five million RMB could be additionally consumed due to early start of Grade III response in August, according to the investigation by GZ CDC. However, around 95,600 RMB could be saved annually if one dengue case was prevented in Guangzhou [32]. Based on our estimates of the averted dengue cases in 2019, around 94,357,200 RMB would be saved due to early start of Grade III response. That is, early start of Grade III

**Table 1. Associations between the logarithm transformation of MOI, early start of Grade III response, and dengue incidence in Model 3.**

| Variable | *RR* | (95% CI) |
| --- | --- | --- |
| log (MOI) | 1.36 | (1.01–1.84) |
| Early start of Grade III response | 0.68 | (0.52–0.90) |

Abbreviations: *RR*, relative risk; 95% CI, 95% confidence interval; log(MOI), logarithm transformation of mosquito ovitrap index.

response could save around 90 million RMB (i.e. 12.89 million USD) in 2019. In addition, results for comparing with no Grade III response suggested that the effect of Grade III response starting in August was much more significant than that commencing in September, which also highlighted the importance of early dengue control. Future studies on comprehensive cost-effectiveness analysis of the Grade III response initiating in August or even earlier before the peak of mosquito vector density occurs can inform suitable public health interventions for dengue prevention and mitigation. The experience in Guangzhou, especially the effective integrated intervention strategy with the cross-departmental collaboration and supervision from senior officials could also be beneficial for other locations with dengue epidemics.

Vector control is of great importance in preventing and mitigating dengue burden [10, 11]. A previous study conducted in Sri Lanka revealed that mosquito vector density could be a mediator of intensified dengue control measures and dengue incidence [15]. Interestingly, we found that mosquito vector density could also be a mediator of early start of Grade III response and dengue incidence. Killing adult mosquito vectors and eliminating mosquito breeding grounds are counterparts of the integrated intervention measures included in the Grade III response to dengue. Therefore, the Grade III response can contribute to a decline in mosquito vector density. Early start of Grade III response reduced the duration of potential exposure to infected mosquito vectors for local residents and subsequently mitigated the dengue burden. Furthermore, it was observed that the effect of bringing forward the start time of Grade III response one month ahead remained statistically significant, after controlling for the effect of the logarithm transformation of MOI (Table 1). The findings implied that early start of Grade III response may have an impact on dengue incidence not only through reducing mosquito vector density, but also in other ways, such as controlling imported dengue cases, strengthening social preparedness and raising public awareness against dengue fever.

Previous studies have showed that dengue incidence was positively associated with household index, container index, and Breteau index in Vietnam [33] and Guangzhou, China [34]. In the present study, we found a positive association between MOI and dengue incidence. Undoubtedly, a higher mosquito vector density could indicate an elevation in the number of mosquitoes infected by dengue viruses. In this situation, the population could be exposed to more infected mosquitoes and therefore at a higher risk of dengue infection. These findings highlighted the importance of reducing mosquito vector density for dengue control. The result of the association between the index of mosquito vector density and dengue incidence can be used for constructing the predictive model for dengue disease burden [4].

The current study had several limitations. First, the number of reported dengue cases was an underestimate of dengue infections in Guangzhou, since almost all reported cases were hospitalized and many asymptomatic people infected with dengue were not identified or reported [2]. Consequently, the number of prevented cases associated with early start of Grade III response would be underestimated. Second, we analyzed 3-year data, while the modification of the start time of Grade III response was observed only for one year. Cumulative data for different start time of response are required to identify an optimal stating time of Grade III response. In addition, because of data availability, the impact of early start of the Grade III response on mosquito vector density and the association between the logarithm transformation of MOI and dengue incidence were assessed on a weekly basis instead of a daily scale, leading to less data points. Next, the intensity of intervention for dengue control might fluctuate during the period of Grade III response. However, the intervention intensity of Grade III response should be within a certain range and should be significantly stronger than that of Grade VI-IV responses given the high degree of government involvement [28]. Since it is extremely difficult to quantify the intervention intensity, we only used a dummy variable indicating the period of Grade III response in 2009 in the model to examine the average effect of

early start of Grade III response on dengue incidence. We explored whether mosquito vector density was a mediator by using Baron and Kenny's approach instead of structural equation modeling (SEM). Although SEM is a powerful technique to distinguish the direct and indirect effects in causal relationships, unfortunately, the techniques currently available for SEM are not applicable to examine the lagged effects for such non-Gaussian time-series data in this study. Further, we considered population, GDP and green coverage when comparing the dengue incidence among Guangzhou, Foshan and Zhongshan, while other variables such as a greater degree of population mobility may lead to higher dengue incidence in Guangzhou, which may be likely to underestimate the impacts of Grade III response.

In conclusion, bringing forward the starting time of Grade III response one month ahead to August have effectively reduced the dengue burden in Guangzhou, China, which might be partially through controlling the mosquito vector density. Our findings have important public health implications on the dengue control interventions for Guangzhou and other locations with dengue epidemics.

## Supporting information

**S1 Fig. Histogram of rainfall.**
(TIF)

**S2 Fig. Models used to test whether mosquito vector density was a mediator between early start of Grade III response and dengue incidence.**
(TIF)

**S3 Fig. Combinations of time lags between climatic variables and mosquito vector density in Model 2 and the lag between logarithm transformation of MOI and dengue incidence in Model 3.** Six combinations of time lags (C1-C6) were shown in the dashed line rectangle. H→M, T→M, and M→D represent the time lags between relative humidity and mosquito vector density, between temperature and mosquito vector density, between mosquito vector density and dengue incidence, respectively.
(TIF)

**S4 Fig. Time-series of monthly number of reported dengue cases, average mosquito ovitrap index (MOI), temperature, absolute humidity, and rainfall in Guangzhou, China, during 2017–2019.**
(TIF)

**S5 Fig. Cumulative relative risk of dengue across temperature and relative humidity.** (A) Cumulative relative risk of dengue across temperature; (B) Cumulative relative risk of dengue across relative humidity. Medians of temperature and relative humidity were treated as the reference levels for the two variables. Grey regions indicate the corresponding 95% confidence intervals.
(TIF)

**S6 Fig. Residual analysis of the models used in the main analysis.** (A) Histogram of residuals of Model 1; (B) Histogram of residuals of Model 2; (C) Histogram of residuals of Model 3; (D) Histogram of residuals of Model 4; (E) Scatter plot of residuals versus predicted daily number of reported dengue cases of Model 1; (F) Scatter plot of residuals versus predicted number of positive ovitraps for adult and larval *Aedes albopictus* of Model 2; (G) Scatter plot of residuals versus predicted weekly number of reported dengue cases of Model 3; (H) Scatter plot of residuals versus predicted weekly number of reported dengue cases of Model 4; (I) Partial autocorrelation function (PACF) of residuals of Model 1; (J) Partial autocorrelation function of

residuals of Model 2; (K) Partial autocorrelation function of residuals of Model 3; (L) Partial autocorrelation function of residuals of Model 4.
(TIF)

**S7 Fig. Relative risk of dengue due to bringing forward the starting time of Grade III response from September to August in 2019 in Guangzhou over 0–21 days.** Grey regions indicate the corresponding 95% confidence intervals.
(TIF)

**S1 Table. Values of Quasi Akaike Information Criterion (QAIC) corresponding to the models which incorporate natural cubic splines of temperature or relative humidity with different degrees of freedom.**
(DOCX)

**S2 Table. Values of Quasi Akaike Information Criterion (QAIC) corresponding to the models which incorporate the cross-basis of an indicator variable of the Grade III response in 2019 and time lag with different maximum lags.**
(DOCX)

**S3 Table. Estimates of averted number of dengue cases due to early start of Grade III response in the sensitivity analysis.**
(DOCX)

**S4 Table. The effect of early start of Grade III response on the rate of positive ovitraps for adult and larval *Aedes albopictus* in the sensitivity analysis.**
(DOCX)

**S5 Table. The effect of the logarithm transformation of mosquito ovitrap index (MOI) on dengue incidence in the sensitivity analysis.**
(DOCX)

**S1 Text. Events related to dengue of six grades and the corresponding responses.**
(DOCX)

**S2 Text. The safety level of mosquito vector density.**
(DOCX)

**S3 Text. Model 1.**
(DOCX)

**S4 Text. Model 2 and Model 3.**
(DOCX)

**S5 Text. Sensitivity analysis.**
(DOCX)

## Acknowledgments

We thank Chen Shi for technique support and staff members of Guangzhou CDC for administrative work and data collection.

## Author Contributions

**Conceptualization:** Lei Luo, Chun-Quan Ou.

**Data curation:** Wen-Hui Liu, Yuan Liu.

**Formal analysis:** Li Li.

**Funding acquisition:** Chun-Quan Ou.

**Investigation:** Li Li, Wen-Hui Liu, Lei Luo.

**Methodology:** Li Li, Xiao-Guang Chen, Lei Luo, Chun-Quan Ou.

**Project administration:** Wen-Hui Liu.

**Resources:** Wen-Hui Liu, Zhou-Bin Zhang, Yuan Liu, Lei Luo.

**Software:** Li Li, Chun-Quan Ou.

**Supervision:** Zhou-Bin Zhang, Xiao-Guang Chen, Chun-Quan Ou.

**Validation:** Wen-Hui Liu, Chun-Quan Ou.

**Visualization:** Li Li, Chun-Quan Ou.

**Writing – original draft:** Li Li, Wen-Hui Liu, Zhou-Bin Zhang.

**Writing – review & editing:** Li Li, Wen-Hui Liu, Zhou-Bin Zhang, Yuan Liu, Xiao-Guang Chen, Lei Luo, Chun-Quan Ou.

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
