## [Decision Letter · Decision Letter 0]

4 May 2020

Dear Dr. Ou,

Thank you very much for submitting your manuscript "The effectiveness of early start of Grade III response to dengue in Guangzhou, China: a population-based interrupted time-series study" for consideration at PLOS Neglected Tropical Diseases. As with all papers reviewed by the journal, your manuscript was reviewed by members of the editorial board and by several independent reviewers. In light of the reviews (below this email), we would like to invite the resubmission of a significantly-revised version that takes into account the reviewers' comments. 

We cannot make any decision about publication until we have seen the revised manuscript and your response to the reviewers' comments. Your revised manuscript is also likely to be sent to reviewers for further evaluation.

Sincerely,

Elvina Viennet, PhD

Deputy Editor

David Harley

Deputy Editor

Reviewer's Responses to Questions

**Key Review Criteria Required for Acceptance?**

**Methods**

-Are the objectives of the study clearly articulated with a clear testable hypothesis stated?

-Is the study design appropriate to address the stated objectives?

-Is the population clearly described and appropriate for the hypothesis being tested?

-Is the sample size sufficient to ensure adequate power to address the hypothesis being tested?

-Were correct statistical analysis used to support conclusions?

-Are there concerns about ethical or regulatory requirements being met?

Reviewer #1: (No Response)

Reviewer #2: 1. The response for the dengue includes six grades, which grade is chosen based on the severity of event. The question is whether the grade will be upgraded to a higher grade when the cases are increasing during an outbreak. For example, the cases were relatively lower when Grade III was launched in 2019, whether the intervention was strengthened with cases increasing. As the authors mentioned in the paper, the responses are comprised mainly of vector surveillance, vector control, raising public awareness etc., which are not quantified measurements, how to sustain the consistency of the intervention during an outbreak, and the comparability among 2017, 2018 and 2019. 

2. The study indicated mosquito density was a mediator of intensified dengue control measures, climatic factors and dengue incidence. Therefore, it may be not appropriate that mosquito density, and climatic factors and dengue control measures were recognized as the independent variables simultaneously to interpret dengue incidence as in model 3, but path analysis or structural equation model can.

3. It’ known that rainfall influences the mosquito density and then influences dengue incidence. I notice that rainfall was not included into the model based on the study’s statistical strategy. Daily data on rainfall were downloaded, why rainfall was arbitrarily categorized into three levels not as the continuous variable was included in the model. 

4. Was lag (Intt, 10-21) or lag (Intt, 10-22) included in model 1?

5. The data on mosquito density was collected once a moth from Jan to Mar, but once a week from Jul to Nov. The variable was included into the model using weekly data, how was the monthly data manipulated?

**Results**

-Does the analysis presented match the analysis plan?

-Are the results clearly and completely presented?

-Are the figures (Tables, Images) of sufficient quality for clarity?

Reviewer #1: (No Response)

Reviewer #2: 1. The results of sensitivity analysis should be submitted in the Sup. Materials. 

2. The lags of climatic factors were selected based on QAIC. The exposure-response association between climatic factors and dengue incidence should be shown, which is helpful for reader to assess the model, not only using residuals.

**Conclusions**

-Are the conclusions supported by the data presented?

-Are the limitations of analysis clearly described?

-Do the authors discuss how these data can be helpful to advance our understanding of the topic under study?

-Is public health relevance addressed?

Reviewer #1: (No Response)

Reviewer #2: The study used an interrupted time series method to explore the effectiveness of early start of grade III response to dengue. The intervention of response as binary variable, yes or no, was entered into the model. However, i still have concerns on the consistency of the interventions. Responses are separated into six grades, but the core may be the same, what difference may be the intensity. How is the response measured?

**Editorial and Data Presentation Modifications?**

Reviewer #1: (No Response)

Reviewer #2: (No Response)

**Summary and General Comments**

Reviewer #1: 1. Up to date, dengue is not endemic in China. So dengue cases consist of imported cases and local cases in China. I recommend the authors divide dengue cases into imported cases and local cases prior to analysis. Grade III response mainly reduce the number of local cases. Of note, the number of local cases not only influenced by Grade III response but also influenced by the number of imported cases.

2. Grade III response can reduce the number of dengue cases. First, Grade III response can reduce mosquito density which can indirect reduce the transmission of dengue viruses. Second, Grade III response may change the activities of humans or improve humans ability of preventing dengue which direct reduce the number of dengue. SEM should be conducted to indentify direct and indirect impacts of Grade III response.

Reviewer #2: The aim of the study is to evaluate the effectiveness of early Grade III response to dengue in the high risk area of China, Guangzhou, which is interesting and is benefit for the local health authority and government. The study did a lot of analyses, and the main results were shown. I think other results should also be submitted in Sup. Materials to support the conclusion.

PLOS authors have the option to publish the peer review history of their article (what does this mean?). If published, this will include your full peer review and any attached files.

Reviewer #1: No

Reviewer #2: No
---

## [Editor Report · Decision Letter 1]

30 Jun 2020

Dear Dr. Ou,

We are pleased to inform you that your manuscript 'The effectiveness of early start of Grade III response to dengue in Guangzhou, China: a population-based interrupted time-series study' has been provisionally accepted for publication in PLOS Neglected Tropical Diseases.

Best regards,

Elvina Viennet, PhD

Deputy Editor

David Harley

Deputy Editor

---

## [Editor Report · Acceptance letter]

29 Jul 2020

Dear Dr. Ou,

We are delighted to inform you that your manuscript, "The effectiveness of early start of Grade III response to dengue in Guangzhou, China: a population-based interrupted time-series study," has been formally accepted for publication in PLOS Neglected Tropical Diseases.

Best regards,

Shaden Kamhawi

co-Editor-in-Chief

Paul Brindley

co-Editor-in-Chief
